# Pharmacological Modulators of Tau Aggregation and Spreading

**DOI:** 10.3390/brainsci10110858

**Published:** 2020-11-13

**Authors:** Antonio Dominguez-Meijide, Eftychia Vasili, Tiago Fleming Outeiro

**Affiliations:** 1Department of Experimental Neurodegeneration, Center for Biostructural Imaging of Neurodegeneration, University Medical Center Goettingen, 37073 Goettingen, Germany; antonio.meijide@usc.es (A.D.-M.); evassili@gmail.com (E.V.); 2Laboratory of Neuroanatomy and Experimental Neurology, Dept. of Morphological Sciences, CIMUS, IDIS, University of Santiago de Compostela, 15782 Santiago de Compostela, Spain; 3Max Planck Institute for Experimental Medicine, 37075 Goettingen, Germany; 4Translational and Clinical Research Institute, Faculty of Medical Sciences, Newcastle University, Framlington Place, Newcastle Upon Tyne NE2 4HH, UK

**Keywords:** tau, tauopathies, Alzheimer’s disease, therapies

## Abstract

Tauopathies are neurodegenerative disorders characterized by the deposition of aggregates composed of abnormal tau protein in the brain. Additionally, misfolded forms of tau can propagate from cell to cell and throughout the brain. This process is thought to lead to the templated misfolding of the native forms of tau, and thereby, to the formation of newer toxic aggregates, thereby propagating the disease. Therefore, modulation of the processes that lead to tau aggregation and spreading is of utmost importance in the fight against tauopathies. In recent years, several molecules have been developed for the modulation of tau aggregation and spreading. In this review, we discuss the processes of tau aggregation and spreading and highlight selected chemicals developed for the modulation of these processes, their usefulness, and putative mechanisms of action. Ultimately, a stronger understanding of the molecular mechanisms involved, and the properties of the substances developed to modulate them, will lead to the development of safer and better strategies for the treatment of tauopathies.

## 1. Introduction

Tauopathies are neurodegenerative disorders characterized by the deposition of abnormal microtubule-associated protein tau (MAPT) in the brain. Tau is a tubulin-associated unit, and tauopathies are a group of clinically, morphologically, and biochemically heterogeneous disorders closely related with dementia [1,2,3]. The spectrum of tau pathologies includes neuropathological phenotypes like Pick’s disease (PiD), progressive supranuclear palsy (PSP), corticobasal degeneration (CBD), argyrophilic grain disease (AGD), primary age-related tauopathy (PART), formerly known as neurofibrillary tangle-only dementia (NFT-dementia), and a recently characterized entity called globular glial tauopathy (GGT) [2,4,5,6,7,8,9]. The neuropathological phenotypes of tauopathies are distinguished on the basis of the involvement of different anatomical areas, cell types and presence of distinct isoforms of tau in the pathological deposits. Nonetheless, the high degree of clinical overlap between different tauopathies limits the specificity of the clinical diagnosis [10]. Additionally, based on whether tau pathology is considered the major contributing factor to neurodegeneration or associated with other pathologies, tauopathies can be divided into primary and secondary [11,12]. PiD, PSP, CBD, AGD, PART, and GGT are primary tauopathies, while Alzheimer’s disease (AD) and chronic traumatic encephalopathy (CTE) are secondary tauopathies [2,8,13,14,15,16,17,18]. Another way to classify tauopathies is based on which isoform of tau is the main one present and, in this regard, tauopathies can be divided into 3R and 4R tauopathies [19].

Regardless of the precise type of pathology, one of the main features is the presence of tau aggregates. As such, aggregation of tau is characteristic of several neurodegenerative diseases. However, tau aggregates are also observed in PART [8,20,21] patients, and these show no obvious cognitive impairment [22].

In this review, we discuss the topic of tau aggregation and spreading and the prospect of pharmacological modulation of these processes.

## 2. Tau Physiology

Tau is a microtubule-associated protein (MAP) mainly expressed in the central and peripheral nervous systems [23] and is encoded by the *MAPT* gene on chromosome 17q21.31 [24]. Tau is a hydrophilic, highly soluble, natively unfolded protein predominantly present in the cytosol and is primarily localized in axons [23,25,26,27], where it promotes the formation of axonal microtubules and stabilizes them by binding at the interface between tubulin heterodimers and drives axonal outgrowth and neuronal plasticity [25,28,29]. The binding and stabilization of microtubules requires the C-terminal region, which regulates the ability of tau to induce microtubule polymerization and its interaction with the plasma membrane [30,31,32,33]. Physiologically, more than 80% of tau is bound to microtubules [24] in an important dynamic process involved in the regulation of neuronal morphogenesis and differentiation. This is mediated by the interaction with the motor proteins kinesin and dynein, thereby regulating neuritic plasticity, axon outgrowth/elongation, and axonal cargo transport to the presynaptic terminal [23,26,27,34,35,36,37,38,39,40].

The microtubule binding domains of tau contain a number of lysine residues, of which positive charges drive tau to bind negatively charged microtubules [41].

Alternative splicing, specifically of exons 2, 3, and 10, generates six known tau isoforms in the adult human brain, ranging from 352 to 441 amino acids [24,40]. Splicing is tissue specific and is developmentally regulated [42,43,44].

Three repeat tau isoforms bind microtubules less effectively than isoforms with four repeats, probably due to the presence of the interrepeat sequence between the first and second microtubule binding domains, which is unique to 4R isoforms [45].

Besides alternative splicing, tau can undergo several posttranslational modifications (PTMs), such as phosphorylation, acetylation, methylation, glycation, isomerization, O-GlcNAcylation, nitration, sumoylation, glycosylation, ubiquitination, and truncation, creating a large heterogeneity of tau molecules that influences tau functions [46,47,48]. Several of these PTMs impact the localization and the propensity for the aggregation of tau.

## 3. Aggregation and Characteristics of Pathological Tau

Thus far, over 50 mutations have been confirmed in the *MAPT* gene. These are classified as missense and splicing mutations and mostly are associated with frontotemporal dementia (FTD), with parkinsonism linked to chromosome 17 (FTDP-17) [49,50,51].

A large group of progressive neurological disorders pathologically defined by the presence of tau inclusions in neuronal and glial cells are collectively known as “tauopathies” [52] and are primarily represented by AD, the most prevalent tauopathy.

In pathological conditions, tau fails to stabilize microtubules and appears as insoluble aggregates which subsequently lead to further tau aggregation, neuronal toxicity, and ultimately neurodegeneration [53,54,55]. Different tau assemblies have been defined including monomers, dimers/trimers, small soluble oligomers, insoluble granular oligomers, filaments, pretangles, neurofibrillary tangles, and ghost tangles [56]. Interestingly, AD brain samples showed a four-fold higher concentration of tau oligomers in comparison to control samples. Studies using atomic force microscopy revealed that tau oligomers consist of dynamic structures that share a spherical-shaped morphology consisting of two or three molecules that are able to turn into β-rich structures with detrimental consequences [57]. Three different types of tau aggregates strongly correlate with neuronal degeneration: the neurofibrillary tangles (NFTs) in neuronal somata, which is the primary cause of neurodegeneration in a number of tauopathies, neuropil threads (NTs) in neuronal dendrites, and neuritic plaques (NPs) [58,59] (Figure 1A). In particular, the density of NFTs correlates fairly well with regional and global aspects of cognitive decline during the progression of AD [60,61]. Although the presence of neurofibrillary tangles in tau inclusions is a critical biomarker for the pathological diagnosis of AD patients, AD is considered as a secondary tauopathy, due to the combined deposits consisting of intracellular NFTs and of extracellular amyloid-β (Aβ) plaques [62,63,64].

The progressive formation of NFTs, consisting predominantly of paired helical filaments, is closely linked to abnormal PTMs of tau proteins [65,66]. In particular, tau proteins isolated from NFTs exhibit a greater degree of abnormal hyperphosphorylation.

As an early pathological event, progressive hyperphosphorylation leads to the dissociation of tau from microtubules, which apparently alters cytoskeleton dynamics and impairs axonal transport, resulting in synaptic dysfunction [67] (Figure 1A). Soluble tau proteins undergo conformational changes that can be key drivers for aggregation to begin [68,69,70,71]. Progressive hyperphosphorylation and increased local tau concentrations in restricted areas promote misfolding and lead to neuropathological alterations in AD brains, particularly correlated with synaptic loss and tangle formation [72,73,74], accompanied by astrogliosis [75,76], and microglial cell activation [76,77,78]. In addition to the deposition of Aβ plaques and neurofibrillary tangles, AD progression is related to cholinergic deficiency, as demonstrated by structural alterations in cholinergic synapses, diminished activity of the acetylcholine-synthesizing enzyme choline acetyltransferase (ChAT), loss of specific subtypes of acetylcholine (ACh) receptors, and the death of ACh-generating neurons, which ultimately impair cholinergic neurotransmission [79,80].

Disrupting tau homeostasis is associated with neurodegeneration, but the precise molecular mechanisms involved are still poorly understood. What is clear is that, in addition to hyperphosphorylation, a variety of factors such as point mutations, truncation, or interaction with binding partners affects tau aggregation in the brain [70,81,82,83,84,85,86].

Tau aggregation can be accelerated by polyanions [87,88]. In vitro, this can be achieved with sulfated glycosaminoglycans (heparin), nucleic acids, acidic peptides, micelles of arachidonic acid, or even carboxylated microbeads [87,88,89,90,91,92]. Additionally, posttranslational modifications, of which phosphorylation is a chief representative, may affect tau aggregation, [93]. In fact, abnormally hyperphosphorylated tau isolated from human AD brains can self-assemble into PHFs in vitro [93]. Truncation can also affect aggregation, as tau fragments that contain the repeat domain have a higher tendency for aggregation [94], a process that is related with pathology [95]. Regardless of the aggregation process, these aggregates can spread throughout the brain, expanding the pathology [96,97]. Additionally, in vitro aggregation studies suggest that the two hexapeptide sequence motifs (VQIINK and VQIVYK) at the beginning of R2 and R3 consist of the core region with a high predicted *β*-structure potential that is crucial for PHF assembly [98,99,100,101]. Especially, the motif VQIVYK forms fibrils composed of steric ‘zippers’ of two tightly interdigitated *β*-sheets leading to aggregation [101,102,103]. The formation of these zippers allows stacking into *β*-sheets that can interdigitate [104]. The disruption of these motifs reduces the tendency for tau to aggregate and, in contrast, strengthening the *β*-structure with certain mutations (for instance, ΔK280 or P301L), accelerates tau aggregation, in vitro and in vivo [104,105,106,107,108].

The self-oligomerization of tau in vitro can be driven by heparin in a process involving two different types of dimers (cysteine dependent or cysteine independent dimers) that is mediated by intermolecular disulfide crosslinking along with PHF hexapeptide [109]. These interactions force the formation of granular and fibrillar tau aggregates. Importantly, atomic force microscopy (AFM) analysis of material derived from AD brains revealed that the amount of granular tau aggregates was elevated in the prefrontal cortex of Braak stage I cases compared to that of Braak stage 0. On the basis of this observation, granular tau species likely precede the formation of PHF and, therefore, may possibly be used as a candidate marker for the diagnosis of certain tauopathies [110].

Tau PTMs generate modified forms of monomeric tau and may, eventually, induce conformational changes that promote aggregation. This does not rule out the possibility that some of those modified monomeric forms may themselves be neurotoxic [111,112,113,114].

## 4. Spreading of Tau Pathology

Many recent studies demonstrated that the progressive accumulation of tau inclusions in specific brain regions in AD and other tauopathies can be explained by the self-propagation of aggregated tau between synaptically connected neurons [96,115,116,117,118], favoring the hypothesis of a prion-like mechanism for the transmission of tau pathology [119,120,121]. This inter-neuronal propagation of abnormal tau, often referred to as “spreading”, appears to occur along neuroanatomically connected areas and requires a continuous repeating process of release/secretion of soluble or aggregated tau from neurons or glial cells, uptake by neighboring recipient cells, and the seeding of intracellular aggregation in the recipient cells [96,97,122,123] (Figure 1B). In agreement with the strain hypothesis, the cell-to-cell transmission seems to be affected by the different conformations of the released and internalized tau protein species [124,125,126,127,128]. According to this concept, different tau strains show distinct seeding capacity upon interaction with endogenous protein [129,130]. Several studies revealed that the spreading of tau pathology could occur via synaptic and non-synaptic mechanisms and that tau species are internalized and transported both anterogradely and retrogradely along neuronal networks [70,131,132].

As we previously reviewed in detail [133], tau can be released by dying/dead cells, secreted by direct translocation across the plasma membrane [42], released by exosomes or other types of extracellular vesicles [29], or transmitted through intercellular cytoplasmic bridges composed of F-actin, known as tunneling nanotubes [43] (Figure 1B). Notably, the release of endogenous tau from neurons might be a physiological process mediated by neuronal activity and is likely to occur in the absence of cell death through a pre-synaptic mechanism. Such release is thought not to be connected with the propagation of tau, indicating that secretion might normally be a regulated process that becomes disrupted in diseased brains [25,134]. Increased levels of tau were also detected in the cerebrospinal fluid (CSF) and in the brain interstitial fluid (ISF) of wild-type and transgenic mice as well as in healthy and AD individuals [135,136,137,138,139]. In a recent study, the tau spreading hypothesis was investigated using positron emission tomography (PET) in human brains. This study suggested that tau is transmitted from cell to cell, mainly through communicating neurons and not through the extracellular space [140].

During AD progression, tau pathology follows a hierarchical pattern of accumulation between anatomically connected brain regions, starting from the transentorhinal cortex, from where it spreads to the hippocampus and neocortex [141]. These findings are further supported by in vivo studies showing that intracerebral inoculation of brain homogenate from mice with filamentous tau pathology induces the progressive development of aggregated hyperphosphorylated tau protein in transgenic mice, expressing wild-type tau, which normally do not show tau aggregates. Furthermore, over time, tau deposition follows a predictable spreading pattern among neighboring brain regions to the injection sites or to each other [122]. Similarly, the intracerebral injection of AD brain-derived tau aggregates into normal C57BL/6 mice can induce cerebral amyloidosis and tau pathology propagation [97]. Using the same approach, the injection of synthetic preformed tau fibrils (pffs) in young asymptomatic PS19 mice expressing mutant human tau (P301S) leads to a rapid induction of NFT-like tau aggregates as well as a time-dependent propagation of tau pathology from injected sites to connected brain regions [142,143], demonstrating overall a template-dependent misfolding of the native tau protein.

The restricted overexpression of human tau P301L in the entorhinal cortex results in the development of filamentous tau pathology, spreading to the dentate gyrus of the hippocampus and synaptic destruction, suggesting the propagation to neighboring synaptically connected neurons [144,145]. In this context, several studies showed that the propagation of tau pathology is dependent on synaptic connectivity rather than spatial proximity, further supporting the involvement of trans-synaptic neuronal mechanisms [131,142,146]. On the other hand, the reduction of tau endogenous levels seems to be protective against neurotoxicity and prevents behavioral deficits in transgenic mice [147,148,149], although overexpressed tau propagates to synaptically connected neurons [126]. Notably, tau derived from AD patients with Aβ plaque pathology appears to be more seeding-competent than tau isolated from cases without Aβ plaques [150]. Furthermore, tau from AD patients is phosphorylated and seed-competent [132,151] and can also be found in brain regions with no extensive tau pathology, confirming the spreading of tau through synaptically connected neurons [152].

Despite numerous compelling studies, the mechanisms that trigger the initial conversion of physiologically soluble proteins into pathogenic polymers remain unresolved. We posit that a better understanding of the underlying mechanisms may lead to the development of novel therapeutic targets.

The neuronal internalization of tau protein is also part of a physiological process, with both monomeric and aggregated species entering neurons through clathrin-mediated endocytosis [73,78], by binding the cell surface HSPGs [66] or by uptake through bulk endocytosis [70] (Figure 1B). Microglia and astrocytes phagocytose extracellular tau as part of the clearance of toxic protein aggregated species, and they are thought to contribute to the spreading, due to the fact that are unable to fully degrade such aggregates. As a response, secretion back to the extracellular space contributes to disease progression [37,38,39,67].

Consistent with this idea, prior to the assembly into fibrils, tau forms soluble oligomers that diffuse and are efficiently taken up by surrounding neurons and are able to seed the aggregation of endogenous tau and cause synaptotoxicity in healthy neurons [70,153,154]. However, it remains unclear whether these tau conformers constitute the primary neurotoxic core that is more prone to propagation throughout the nervous system.

## 5. Pharmacological Modulators of Tau Aggregation

Since the 1990s, several treatments have been tried for different tauopathies. However, several of the strategies developed were only symptomatic, such as cholinesterase inhibitors [155,156] and NMDA-receptor antagonists such as memantine [157]. These strategies are only aimed at ameliorating some of the symptoms and not at modulating disease progression [155,156,157]. Therefore, in recent years, several strategies targeting Aβ, amyloid precursor protein (APP), and tau have been developed and entered clinical trials [155,158,159,160]. Thus far, all clinical trials targeting Aβ have produced negative or somewhat disappointing results. Therefore, there is growing interest in targeting tau as a possible alternative [161].

Different tau-based strategies have been considered, such as microtubule stabilization, immunotherapy, O-GlcNAc inhibition (O-GlcNAcases), and tau aggregation inhibition. Among them, inhibition of tau aggregation is the most widely investigated strategy in AD [162], as the substantial increase in bulk tau levels that accompanies lesion formation results primarily from the accumulation of insoluble tau aggregates [163,164].

Two different pharmacological strategies aiming at inhibiting tau aggregation have been developed. One consists of the direct binding to tau, keeping it in an interaction-incompetent conformation, thereby hampering its aggregation [163,165]. This strategy poses some difficulties because tau is an intrinsically disordered protein. Therefore, the rational drug design strategy that has been used successfully since the 1980s cannot be used, as it relies on knowledge of the three-dimensional structure of the target protein, for the design of ligands (usually inhibitors) with the aid of computational tools [166,167]. The other strategy is based on interactions (that do not need to be direct binding) that promote the stabilization of non-toxic species [163,165,168,169].

Two six-residue segments, VQIINK at the start of repeat 2 and VQIVYK at the start of repeat 3, drive the formation of amyloid aggregates of tau [101,165,170]. Hence, the first therapeutic strategy should focus on targeting those sequences by the use of covalent inhibitors that can either covalently modify tau directly or foster formation of covalent bonds within or between tau proteins to yield aggregation-incompetent products (Figure 2A and Figure 3). Covalent inhibitors can attack any or all species in an aggregation pathway, but appear to be especially efficacious modifiers of tau monomer, from which all aggregated species ultimately derive [163]. Additionally, inhibitors should be able to cross the blood-brain barrier [163].

The first covalent inhibitor used was methylthioninium chloride (methylene blue, MB) [171], a phenotyacine dye first developed in 1876. MB binds to the repeat domain of tau, blocking tau-tau interactions during paired helical filament (PHF) formation [105]. This compound has a potentially broad pharmacology, including antibacterial properties, inhibition of microtubule assembly, inhibition of butyrilcholinesterase, inhibition of noradrenalin re-uptake, increase in serotonin extracellular levels, and modulation of AMPA/kainate and NMDA-type ionotropic glutamate receptors [7,172,173,174,175,176,177,178,179]. Additionally, it can attenuate tauopathy by induction of autophagy, by inhibition of Hsp70 ATPase activity, and by cysteine oxidation [180,181,182]. Furthermore, in P301L mutant mice, MB reduces abnormal tau accumulation [183]. It was the first compound that underwent clinical trials as a tau aggregation inhibitor [184], but it was discarded in phase III. Atomic force microscopy studies revealed that MB reduces the number of tau fibrils but increases the number of granular tau oligomers, which has been proposed as an explanation for its failure [185].

Aminothienopyridazines are compounds related to MB which act by accelerating disulfide bond formation inside and between tau molecules, via cysteine oxidation [182]. Several modifications have been made to this family of compounds, leading to the development of new modulators of tau aggregation [186]. Among these modulators, leuco-methylthioninium bis(hydromethanesulfonate) (LMTM) [187] reached phase III clinical trials for FTD [188,189]. However, it did not show benefit when tested at two doses in participants with mild-to-moderate AD, for unknown reasons [189,190]. Additionally, the effects of this inhibitor are affected by anticholinesterase, as LMTM increases hippocampal acetylcholine levels [187]. Another compound from this family that underwent clinical trials was hydromethylthionine [191], reaching phase III, where it failed to reach the primary efficacy endpoints in terms of attenuating the rate of progression of the disease at doses in the range of 150–250 mg daily [191]. Interestingly, compounds from this family can be used as imaging probes for different tauopathies [192].

Other covalent inhibitors include: oleocanthal, a natural aldehyde with anti-inflammatory properties present in olive oil [193] that reacts with tau lysines, especially lysine 311, reducing filament formation [194,195]; cinnamaldehyde, which blocks tau aggregation by undergoing nucleophilic attack by the cysteine residues of tau [196], a specific mechanism of tau aggregation inhibition common with other aldehydes, such as the *Aspergillus nidulans* metabolite asperbenzaldehyde [163,197], and several azaphilone derivatives [198]; baicalein, a polyalcohol flavonoid, is oxidized to the quinone form, before acting as a covalent inhibitor [199]. However, covalent inhibitors may interact non-specifically with other proteins, causing off-target effects [163] and, therefore, this type of substances has long been avoided, due to the fear of unspecific modifications and the fear that the haptenization of modified proteins might lead to an immune response [200,201].

The second group of inhibitors consists of non-covalent inhibitors, consisting of structurally and mechanistically diverse molecules [163,202] (Figure 2B and Figure 3). The mechanisms of action are diverse and they can be further classified into several different groups [163]. Due to the similarities and interactions between alpha-synuclein (aSyn) and tau [133,203,204], compounds that have proven to be effective against aSyn aggregation may also modulate tau aggregation. One of these compounds, now undergoing clinical trials, is curcumin [205]. Curcumin is a yellow-orange polyphenol compound found in abundance in the rhizome of the plant *Curcuma longa* [206]. Curcumin affects aSyn, Aβ, and tau aggregation and can also inhibit Aβ production [207,208,209,210]. However, it is unclear whether curcumin or other related ligands can be optimized to interact with a specific molecular target. This is relevant because cross-reactivities with other natively unfolded peptides might occur [211]. Therefore, several derivatives aimed at acting more specifically on tau and/or Aβ have been developed [212,213,214].

Another group of non-covalent inhibitors are molecular tweezers, such as CLR01. Like with curcumin and its derivatives, the mechanism of action of these compounds was first elucidated for aSyn, where they lower the aggregation propensity by increasing the reconfiguration rate, similarly to curcumin [215]. They interact with tau through lysine side chains, inhibiting its aggregation [216]. In vivo studies in mice showed a decrease in Aβ levels and tau burden in animals treated with CLR01 [217].

Similarly to molecular tweezers and curcumin, steric zipper blockers such as Orange-G also interact with tau through its lysine side chains [218]. As the name suggests, the mechanism of action blocks the formation of steric zipper structures common to cross-β-sheet forming peptides [163]. Using X-ray diffraction, it was shown that the blocking of tau aggregation is performed by the aromatic rings of Orange-G, which are packed against apolar side chains of Val309, establishing polar interactions with glutamine and lysine side chains at the edges of the steric zipper [219].

Apart from interacting with lysine chains, curcumin derivatives, molecular tweezers, and steric zipper blockers share another common characteristic, which is that they bind with their long axes parallel to the fiber axis [218,219]. Therefore, it has been proposed that these modulators should be combined in therapeutic cocktails [218].

Other compounds such as cyanine, rhodamine, and triarylmethine derivatives (such as crystal violet) may decrease tau aggregation by stabilizing soluble oligomeric species at the expense of filamentous aggregates [220]. Amongst them, cyanine does not interact with natively unfolded tau monomers and leads to the formation of off-path tau oligomers unable to further elongate [220]. Using structure activity relationship (SAR) analysis, it was observed that they can be more potent than methylene blue [221]. However, they lead to an increase in oligomeric species which results in the formation of PHF and NFTs [111,112,113,114]. Other compounds, such as phthalocyanine tetrasulfonate (PcTS) modulate tau aggregation by targeting the protein into soluble oligomers, thus interfering with filament formation [222]. This has been shown in vitro by NMR spectroscopy, electron paramagnetic resonance, and small-angle X-ray scattering, observing that the soluble tau oligomers contain a dynamic, non-cooperatively stabilized core with a diameter of 30–40 nm that is distinct from the core of tau filaments [222].

Another group of tau aggregation inhibitors are dibenzofuran derivatives, such as usnic acid and its derivatives. Usnic acid is a high-level secondary metabolite in lichen [223] that decreases tau aggregation in vivo and in vitro [224] but, so far, the mechanism of action is still not completely understood. Another metabolite, fulvic acid, is a mixture of different polyphenolic acids produced by humus that decreases heparin-induced tau aggregation in vitro [225,226]. We have also observed that fulvic acid inhibits K18 tau aggregation in vitro and full-length tau in a cell model. Interestingly, fulvic acid seems to disaggregate previously formed tau aggregates in cells, in agreement with findings using heparin-induced tau aggregation.

Other compounds that act, not only inhibiting aggregation but also disaggregating aggregates already formed, are naphtoquinone-tryptophan derivatives such as NQTrp, and its most stable derivative Cl-NQTrp, which significantly disrupted pre-formed fibrillar aggregates of Tau-derived PHF6 (VQIVYK) peptide and full-length tau protein, both in vitro and in a drosophila model [227,228]. These compounds target not only tau but also Aβ and, possibly, aSyn, as it has been observed that mannitol-NQTrp conjugates decrease aSyn aggregation in vitro [229,230].

Additionally, as mentioned above, compounds that can modulate aSyn aggregation have also been used used to modulate tau aggregation. In this regard, we tested Anle138b in a cell model and observed a decrease of tau aggregation [231]. Anle138b is a diphenylpyrazole that blocks aSyn, Aβ and tau aggregation [232,233,234,235,236]. Anle138b seems to be effective against pre-formed tau aggregates [231] and has been shown to ameliorate pathology and metabolic decline in mouse models of tauopathies [235,236]. Furthermore, compounds that inhibit the aggregation of Aβ might also be effective against tau aggregation. One such example is the aforementioned curcumin [237]. Curcumin and other secondary metabolites from plants affect Aβ aggregation, as observed by thioflavin T assay. These substances include rosmarinic acid, gallic acid, salvianolic acid B, luteolin, quercetin, fisetin, myricetin, dihidromyricetin, EGCG, silibinin, oleuropein, rutin, curcumin, crocin, cryptotanshinone, and tabersonine [237]. Among these substances, EGCG also inhibits aSyn aggregation, but shows no effect on tau aggregation in cell culture [231].

There are also modulators that, rather than inhibiting tau aggregation, increase aggregation. The study of these compounds is also interesting because they afford us a handle on the aggregation process as experimental tools. In recent years, several heparin-based in vitro methods to induce tau aggregation have been developed [89,90,238,239,240], but some of these methods lead to the formation of aggregates whose structures differ from those formed in tauopathies [241,242]. Polyanionic substances promote paired helical filament formation and do not strongly affect tau binding to microtubules [107]. Two different mechanisms have been proposed to explain this: a nucleation-dependent polymerization (NDP) [243] and a nucleation-independent mechanism [244].

Recently, it was proposed that tau polymerizes through association with cofactors to form a metastable complex that remains “inert” and reversible, until encountering a relevant seed that can trigger an irreversible transition to β-sheet containing species [245], consistently with the NDP mechanism [246]. This happens not only with heparin but also with nucleic acids [247] and other anionic compounds [248]. It has also been shown that the liquid-liquid phase separation can initiate tau aggregation [249]. However, since heparin-induced tau aggregates are different from real pathological aggregates found in the brains of patients, the use of heparin-induced aggregates as models for aggregation inhibition needs to be interpreted with caution [250].

Additionally, other tau aggregation modulators may act indirectly, by affecting metabolic pathways that may regulate tau expression and aggregation. Tau protein accumulation is regulated by a chaperone system involving Hsp90 and Hsp70 [251]. Hsp90 binds tau, causing a conformational change that allows tau phosphorylation by glycogen synthase kinase (GSK3β), leading to tau aggregation [252,253,254]. Hsp70, on the other hand, has shown to inhibit nucleation and the elongation of tau and sequesters tau aggregates with high affinity in the ΔK280 variant [255]. Therefore, compounds acting on Hsp70, Hsp90, or GSK3β may indirectly modulate tau aggregation. Several such compounds are being developed, including the 1H-pyrrolo [2,3-b]pyridine derivative, B10 [256], which affects tau aggregation by inhibiting GSK3β, the Hsc70/Hsp90 inhibitor 17-AAG and Hsp90 inhibitor KU-32 [257], and Aha1, an activator of Hsp90 that drives the formation of pathological tau aggregates [258]. Substances that act on Hsp70 include MKT-077 and YM-1, rhodocyanines that cause selective death of cancer cells and bind, with low micromolar affinity, to the nucleotide binding domain of ADP- but not ATP-bound Hsp70, stabilizing the ADP-bound state [259].

Other compounds acting indirectly on tau aggregation include modulators of PP2A, PP5, or other kinases (such as ALK), and affect aggregation by modulating its phosphorylation status [260,261,262,263]. As mentioned above, progressive hyperphosphorylation leads to the dissociation of tau from microtubules, promotes misfolding and aggregation and leads to neuropathological alterations [67,68,69,70,71]. Thus, another possible strategy to counteract tau aggregation may be by decreasing hyperphosphorylation by acting on kinases. Amongst these, inhibition of tyrosine kinases has especial relevance, as it leads to the progression of the disease in AD and Parkinson’s disease (PD) [264,265]. In fact, tau has 5 tyrosine residues: 18, 29, 197, 310, and 394. Tyrosines 18, 197, and 394 have been shown to be phosphorylated in AD patients [266]. In recent years, several anticancer drugs have been repurposed for the study of their effects in neurodegenerative diseases. These drugs act on different tyrosine kinases showing effects in different pathologies such as AD, PD, stroke, spinal cord injury, and multiple sclerosis [264,265,267,268,269,270]. Among these drugs nilotinib, dasatinib, vatalanib, and imatinib have shown promising results. Nilotinib is undergoing phase 2 clinical trials, where its safety is being tested [271,272], while dasatinib has been approved for clinical use in senescent cell clearing in the United States since 2006 [273]. Vatalanib can affect VEGF and decreases Aβ accumulation [274], and imatinib shows a broad spectrum of activities that may allow for its future use in several different pathologies [269,275]. Another modulator of phosphorylation that can affect tau aggregation is davunetide, a small peptide whose efficiency has been tested in several in vitro and mouse models [276,277,278] and was tested in a phase III clinical trial for progressive supranuclear palsy, albeit with a negative outcome [279].

TPI-287, a taxane derivative, is another modulator of tau aggregation that may affect cancer cells, as it stabilizes microtubular structures [280]. TPI-287is currently in a phase I clinical study for the treatment of mild to moderate AD and for some other disorders related to the disruption of intracellular transport [158,281].

In recent years, other alternative strategies for modulating tau aggregation have been developed, such as knockdown strategies [282] and immunotherapies [254,283], but these are outside the scope of the present manuscript and should be discussed elsewhere, as they require a dedicated review.

## 6. Pharmacological Modulators of Tau Spreading

Another possible therapeutic strategy in tauopathies is the modulation of tau pathology spreading, assuming there is a causal role between the accumulation of tau and tauopathies. This strategy is based on the hypothesis that tau, as other proteins associated with neurodegeneration, spread in a prion-like manner from neuron to neuron [161]. This has been widely studied in animal models and involves several different mechanisms, as mentioned above. Therefore, several different strategies are being tested to interfere with tau spreading (Figure 2C).

Tau can spread via extracellular vesicles, such as exosomes [284,285]. Therefore, the modulation of exosomes or other extracellular vesicles, for example by mTor1 inhibitors like rapamycin, may interfere with tau spreading, since mTor1 regulates exosome release on the basis of nutrient and growth factor conditions [286]. In addition, on the basis of growth factor conditions, it has been proposed that growth hormone-releasing hormone (GHRH) may modulate the release of neuronal exosomes and, thereby, tau spreading [287]. However, the results obtained so far are not conclusive, and it remains unclear whether GHRH impacts the clearing mechanisms involved in reducing AD pathology in the brain [287]. In addition, exosomes may themselves be used as vehicles to carry different treatments to hamper tau progression or even as diagnostic tools in tauopathies [288].

Tau can interact with HSPGs, which will lead to its internalization [66]. Thus, an approach to modulate tau spreading might be to use of exogenous PG mimetics, including heparin [289]. SN-13 is a heparin-derivative developed from pentasaccharide units that inhibits tau aggregate propagation in a similar way to heparin [290]. Additionally, simple heparin-like oligosaccharides bearing 2-O, 6-O, and N-sulfation can bind strongly to tau oligomers, blocking their internalization in SH-SY5Y cells [289]. Animal studies have shown that heparin-mimetics are promising agents for inhibiting prion protein pathogenesis [291], further supporting the idea that drugs targeting HSPGs might act as modulators of tau spreading.

Tau can also spread through tunneling nanotubes in cultured cells [43,292]. However, it is still unclear whether this process happens also in the human brain [43,292]. So far, there are no pharmacological modulators capable of modulating tau spreading via tunneling nanotubes and, therefore, additional research is needed on this topic [293].

Another mechanism through which tau may spread is by receptor-mediated endocytosis. Tau can be internalized via clathrin-mediated endocytosis, a process mediated by dynamin [294]. This mechanism was associated with the spreading of monomeric tau in iPSCs-derived human neurons [73] and could be modulated by dynasore, a reversible and non-competitive dynamin 1 and dynamin 2 inhibitor [295,296].

Tau uptake also seems to be mediated by M1 and M3 muscarinic receptors [297], G-coupled cholinergic receptors whose activation triggers several different second messenger cascades in neurons [298]. M1 agonists appear to be good candidates as modulators of tau spreading, suggesting additional studies are necessary.

Another mechanism associated with tau internalization is micropinocytosis [66], an actin-driven endocytic process involving the formation of the macropinosome in response to the direct actions of cargo/receptor molecules that coordinate the activity and recruitment of specific effector molecules and subsequently fuse with degradative compartments of the cell [299,300]. Screening of 640 FDA-approved compounds through a cell-based assay lead to the identification of seven inhibitors: uranofin, flubendazole, imipramine, itraconazole, phenoxybenzamine, terfenadine, and vinblastine [301]. Other compounds that act on macropinocytosis are Cytochalasin D, which disrupt several clathrin-independent endocytic processes, including bulk endocytosis/micropinocytosis [302].

Another possibility for modulating tau spreading might be reducing tau levels, either by reducing tau expression, or by promoting its clearance [169]. Therefore, small molecules, immunotherapies, or genetic interventions might be suitable for reducing tau levels and, thereby, tau spreading. The latter two strategies are beyond the scope of the present manuscript and should be discussed separately.

Tau undergoes proteasomal degradation through two different (but not mutually exclusive) pathways, ubiquitin-dependent and ubiquitin-independent [303]. Molecules that modulate protein ubiquitination include TH006, MG132, and QC-01–175 [282,304,305]. TH006 has been shown to regulate tau levels in the mouse brain [304], while QC-01–175 was derived from a PET tracer and optimized to increase tau degradation and has proven effective in primary human neuronal cultures [305]. MG132 indirectly acts on tau ubiquitination by directly biding to tau and keap1 [282], a protein that facilitates ubiquitination by binding to other proteins, anchoring them in the cytoplasm [306].

## 7. Concluding Remarks

In recent years, there has been an increasing number of molecules developed to act on tau oligomerization and aggregation. The first tau aggregation inhibitors developed were based on direct binding to tau. Unfortunately, as tau is an intrinsically disordered protein, traditional structure-activity approaches cannot be performed, thus making the development of direct ligands very difficult. Therefore, alternative approaches have been developed. For example, inhibitors of aSyn aggregation were tested and have proven to be effective against tau aggregation as well. These non-covalent aggregation inhibitors act through diverse mechanisms of action and they can be combined in therapeutic cocktails. Additionally, the development of molecules that promote aggregation is interesting for the study of the aggregation process. Likewise, compounds aimed at modulating oligomerization are also attractive, as they may also enable the interrogation of important biology and constitute potential therapeutic strategies.

Additionally, as tau may spread in a prion-like manner from neuron to neuron, blocking tau spreading is a promising strategy to stop disease progression. In this regard, different compounds are being developed to block specific mechanisms by which tau may spread, including via exosomes, internalization by HSPGs, receptor-mediated endocytosis, and micropinocytosis.

To conclude, impressive advances have been made in the development of molecules with therapeutic potential. While true therapeutic success has not happened yet, failures have also pushed the development of better and safer molecules that enable us to test our hypothesis and continue our quest to develop effective therapies for tauopathies.

## Figures and Tables

**Figure 1 brainsci-10-00858-f001:**
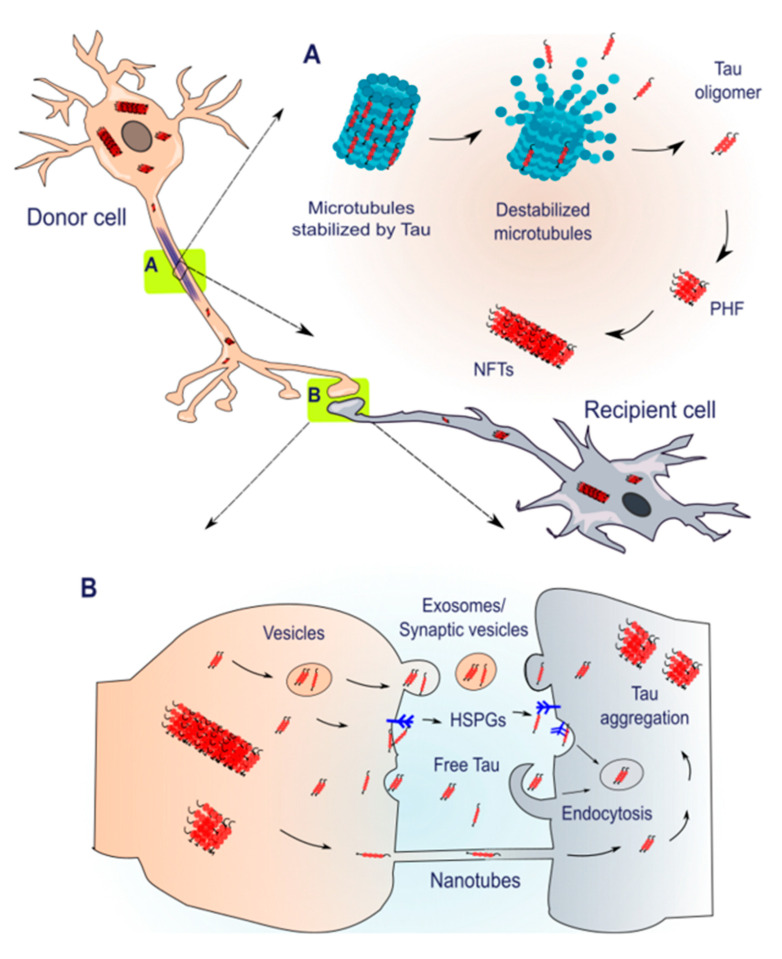
Tau aggregation and spreading. (**A**) Tau stabilizes microtubules and contributes to the maintenance of axonal shape and neuronal morphology. In pathological conditions, tau is subjected to various posttranslational modifications (PTMs) that reduce microtubule binding and, thereby, promote the generation of insoluble tau. Soluble monomers form oligomers which aggregate to generate paired helical filaments (PHFs). These, in turn, assemble to produce neurofibrillary tangles (NFTs). (**B**) Possible mechanisms involved in the cell-to-cell transmission of pathological tau. Transmission may occur via direct translocation across the plasma membrane, via exosomes/synaptic vesicles, by clustering with the plasma membrane and interaction with the cell Heparan sulfate proteoglycans (HSPGs), and through cytoplasmic bridges called tunneling nanotubes. Uptake from the extracellular space can be mediated by binding and internalization with HSPGs or through endocytosis.

**Figure 2 brainsci-10-00858-f002:**
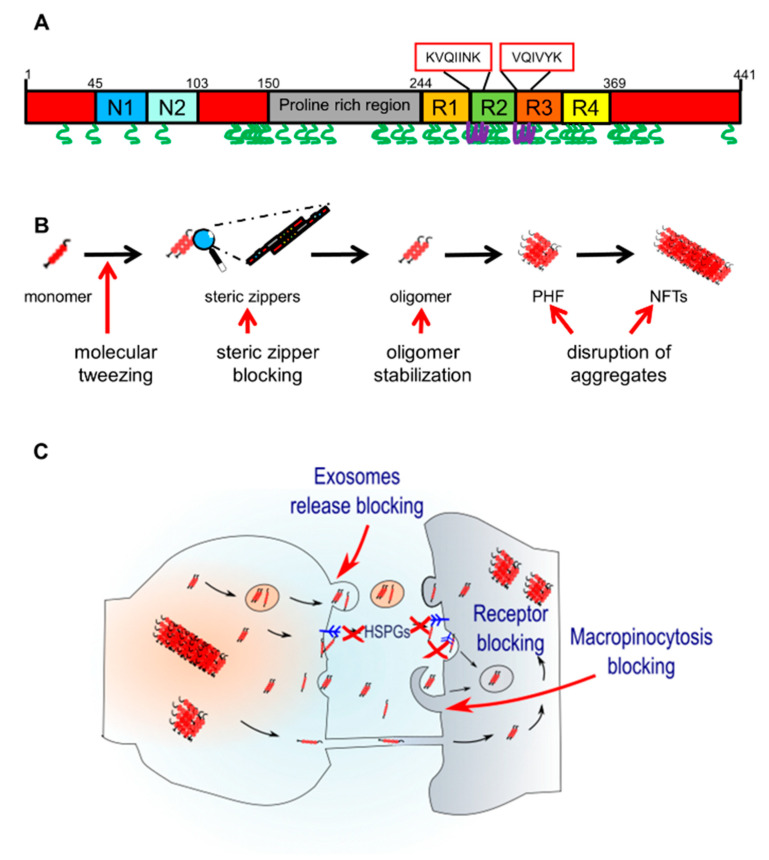
Inhibition of tau aggregation and spreading. (**A**) Sites for direct binding of covalent tau aggregation inhibitors. There are two main groups that bind to the VQIINK and VQIVYK sequences (purple) or to the K residues (green). (**B**) Mechanisms of action of non-covalent tau aggregation inhibitors on tau aggregation. The red arrows point to the different points of inhibition. Molecular tweezers lower aggregation propensity by increasing reconfiguration rate, steric zipper blockers block the formation of the steric zippers structures, and oligomerization stabilizers block the process in the oligomer phase. Finally, PHFs and NFTs can be broken by aggregation disruptors. (**C**) Tau spreading can be inhibited by blocking exosomal release, by blocking tau interaction with HSPG, or by blocking endocytosis, either by blocking the receptors responsible for tau internalization or by blocking micropinocytosis.

**Figure 3 brainsci-10-00858-f003:**
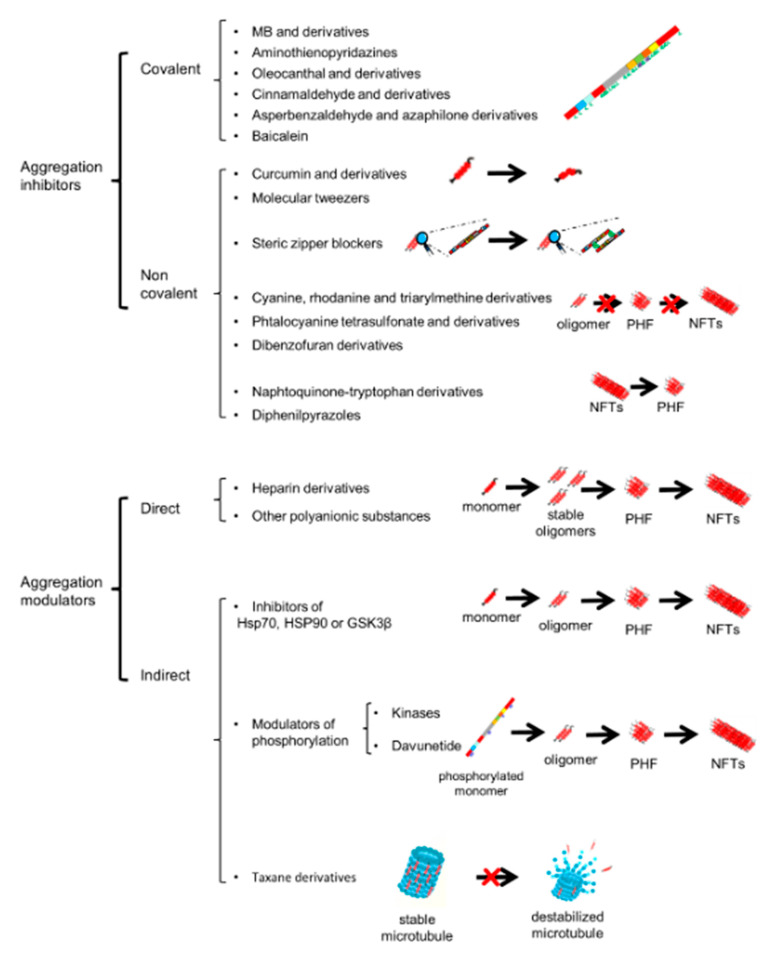
Schematic representation of the main compound groups mentioned in the text and their corresponding main mechanisms of action. Main interaction sites with the protein for covalent inhibitors and steric zipper blockers are shown in green. Tyrosines susceptible of phosphorylation by tyrosine kinases are show in blue.

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
