# Peer review of "Pharmacological Modulators of Tau Aggregation and Spreading"

_brainsci, 2020, doi:10.3390/brainsci10110858_

Round 1

Reviewer 1 Report

The review submitted Dominguez-Meijide and co-authors entitled "Pharmacological modulators of Tau Aggregation and Spreading" is well written and covers several mechanisms that target tauopathies and its different types. It basically covers the active phytochemical components from the herbs as well. Although the review explains in detail the mechanisms involved in tauopathies and its target sites for the treatment strategies, it lacks in the information which might be useful to correlate the Tau protein and its aggregation due to other physiological pathways viz alpha-synuclein, beta-amyloid proteins, tyrosine kinase pathways, etc. The review lacks the consideration and discussion of the ongoing pharmacotherapy for Alzheimer's disease. The review needs to add more information on the existing therapeutic approaches which might be useful information to the readers.

Major comment:

1. The review lacks the information on the kinases which are mainly important in the pathophysiology of AD. For example, Tyrosine kinase, which is one of the main active sites and is responsible for the aggregation of beta-amyloid, tau and alpha-synuclein. It is also responsible for the progression of the disease in patients with any of these pathophysiologies. Tyrosine kinase inhibitors like nilotinib, dasatinib, etc have shown promising results in the treatment of AD-related dementia in patients. Also, it will be worth to add the kinase profiles of these compounds along with the target sites and discuss them in the revision.

2. The authors should add one diagrammatic representation of all the active components covered by them in the review which is showing their main target sites. This will help the readers to understand the review easily since it is too wordy in terms to understand and take the key points in its present format.

Minor:
1. Manuscript needs a thorough check of English grammar and font in terms of a journal style.

2. Line 464-466: On the other hand, development of molecules that can be used as tool compounds that promote aggregation is of also of interest for the study of the aggregation process. Statement not clear. Please reframe the sentence for easy understanding.

Author Response

We want to thank the reviewers for their interest in our work, their time and their valuable comments, that have enabled us to improve this manuscript.

Response to reviewer 1

The review submitted Dominguez-Meijide and co-authors entitled "Pharmacological modulators of Tau Aggregation and Spreading" is well written and covers several mechanisms that target tauopathies and its different types. It basically covers the active phytochemical components from the herbs as well. Although the review explains in detail the mechanisms involved in tauopathies and its target sites for the treatment strategies, it lacks in the information which might be useful to correlate the Tau protein and its aggregation due to other physiological pathways viz alpha-synuclein, beta-amyloid proteins, tyrosine kinase pathways, etc. The review lacks the consideration and discussion of the ongoing pharmacotherapy for Alzheimer's disease. The review needs to add more information on the existing therapeutic approaches which might be useful information to the readers.

We thank the reviewer for this comment. As the reviewer might understand, while the points raised are indeed very interesting, they are not within the scope of the topic of our manuscript. Nevertheless, we added further discussion on the ongoing pharmacotherapy for AD. We have added some sentences on current existing pharmacotherapy for AD after line 236.

Major comment:

  1. The review lacks the information on the kinases which are mainly important in the pathophysiology of AD. For example, Tyrosine kinase, which is one of the main active sites and is responsible for the aggregation of beta-amyloid, tau and alpha-synuclein. It is also responsible for the progression of the disease in patients with any of these pathophysiologies. Tyrosine kinase inhibitors like nilotinib, dasatinib, etc have shown promising results in the treatment of AD-related dementia in patients. Also, it will be worth to add the kinase profiles of these compounds along with the target sites and discuss them in the revision.

We thank the reviewer for this comment. Again, while the information on kinases is interesting, it is outside the scope of the manuscript. In any case, we have added a paragraph on tyrosine kinase and comment on different tyrosine kinase inhibitors, adding their profiles. These changes are in lines 395-409.

  1. The authors should add one diagrammatic representation of all the active components covered by them in the review which is showing their main target sites. This will help the readers to understand the review easily since it is too wordy in terms to understand and take the key points in its present format.

We thank the reviewer for this comment. We have made a new figure (figure 3) that represents the compounds that act on tau aggregation and their main target sites while. The compounds that act on tau spreading were already represented in figure 2 C.

Minor:
1. Manuscript needs a thorough check of English grammar and font in terms of a journal style.

We have revised the English as suggested. The formatting is usually taken care by the journals, so this is something they will verify later.

  1. Line 464-466: On the other hand, development of molecules that can be used as tool compounds that promote aggregation is of also of interest for the study of the aggregation process. Statement not clear. Please reframe the sentence for easy understanding.

We have revised the sentence - now in line 506.

Reviewer 2 Report

The manuscript titled as “Pharmacological modulators of tau aggregation and spreading" by Dominguez-Meijide et al., focus on  processes of tau aggregation and spreading, and highlight selected chemicals developed for the modulation of these processes, their usefulness and putative mechanisms of action.

This work should be of wide interests to most researchers on neuroscience and molecular medicine etc. The review is well structured and annotated with several relevant figures to annotate the key issues of the review. This manuscript is well written, organized and sounds with a good standard of English language. The following points need to be addressed:

Section 3 Aggregation and Characteristics of Pathological Tau  - There are few 2020 publications which emphasizes aggregation and characteristics of pathological tau  (Stanciu et al., 2020, Biomolecules 2020, 10, 40; Stefanescu et al., Biomolecules 2020, 10, 870; Vogel et al., 2020, Nature Communications 11:2612; Mamun et al., 2020, Neural Regen Res; 15:1417-20; Vogel et al., 2020, Biological Psychiatry; 87:808–818) that are not referenced at all in this review. It may have been made available online after the search period for this review but the review does not list search cutoff date so this is unclear. The article and any related information should be reviewed and included in the revisions.

It is necessary to increase the number of references in the last 3 years. Their share should be about half of all references.

Typing error:

Line 356: compounds instead of cocmpounds

Add a space between variant [255] - line 365, brain[281, 282] - line 411.

Author Response

We want to thank the reviewers for their interest in our work, their time and their valuable comments, that have enabled us to improve this manuscript.

Response to reviewer 2

The manuscript titled as “Pharmacological modulators of tau aggregation and spreading" by Dominguez-Meijide et al., focus on  processes of tau aggregation and spreading, and highlight selected chemicals developed for the modulation of these processes, their usefulness and putative mechanisms of action.

This work should be of wide interests to most researchers on neuroscience and molecular medicine etc. The review is well structured and annotated with several relevant figures to annotate the key issues of the review. This manuscript is well written, organized and sounds with a good standard of English language. The following points need to be addressed:

Section 3 Aggregation and Characteristics of Pathological Tau  - There are few 2020 publications which emphasizes aggregation and characteristics of pathological tau  (Stanciu et al., 2020, Biomolecules 2020, 10, 40; Stefanescu et al., Biomolecules 2020, 10, 870; Vogel et al., 2020, Nature Communications 11:2612; Mamun et al., 2020, Neural Regen Res; 15:1417-20; Vogel et al., 2020, Biological Psychiatry; 87:808–818) that are not referenced at all in this review. It may have been made available online after the search period for this review but the review does not list search cutoff date so this is unclear. The article and any related information should be reviewed and included in the revisions.

We thank the reviewer for this suggestion. Indeed, several of those manuscripts were published after we started working on our manuscript. We have now commented on these studies findings and added the corresponding references.

It is necessary to increase the number of references in the last 3 years. Their share should be about half of all references.

There is no rule as to what the percentage of references should be, so we cited studies that we considered relevant. In any case, we added more recent references as suggested.

Typing error:

Line 356: compounds instead of cocmpounds

We have corrected this typo.

Add a space between variant [255] - line 365, brain[281, 282] - line 411.

We have corrected this typo.